# Major Pulmonary Resection for Non-Small Cell Lung Carcinoma during the COVID-19 Pandemic—Single Israeli Center Cross-Sectional Study

**DOI:** 10.3390/jcm11041102

**Published:** 2022-02-19

**Authors:** Michael Peer, Sharbel Azzam, Marina Kolodii, Yaacov Abramov, Ruth Shaylor, Vladimir Verenkin, Nachum Nesher, Idit Matot

**Affiliations:** 1Department of Thoracic Surgery, Tel Aviv Medical Center, Weizman Street 6, Tel Aviv 6423906, Israel; sharbelazzam8@hotmail.com (S.A.); marina05060@gmail.com (M.K.); yyakov@msn.com (Y.A.); nesher61@gmail.com (N.N.); 2Department of Anesthesiology, Tel Aviv Medical Center, Weizman Street 6, Tel Aviv 6423906, Israel; ruth@shaylor.co.uk (R.S.); vladimirv@tlvmc.gov.il (V.V.); iditm@tlvmc.gov.il (I.M.)

**Keywords:** major pulmonary resection, non-small cell lung carcinoma, COVID-19

## Abstract

Background: The highly contagious COVID-19 has created unprecedented challenges in providing care to patients with resectable non-small cell lung carcinoma (NSCLC). Surgical management now needs to consider the risks of malignant disease progression by delaying surgery, and those of COVID-19 transmission to patients and operating room staff. The goal of our study was to describe our experience in providing both emergent and elective surgical procedures for patients with NSCLC during the COVID-19 pandemic in Israel, and to present our point of view regarding the safety of performing lung cancer surgery. Methods: This observational cross-sectional study included all consecutive patients with NSCLC who operated at Tel Aviv Medical Center, a large university-affiliated hospital, from February 2020 through December 2020, during the COVID-19 pandemic in Israel. The patients’ demographics, COVID-19 preoperative screening results, type and side of surgery, pathology results, morbidity and mortality rates, postoperative complications, including pulmonary complications management, and hospital stay were evaluated. Results: Included in the study were 113 patients, 68 males (60.2%) and 45 females (39.8%), with a median age of 68.2 years (range, 41–89). Of these 113 patients, 83 (73.5%) underwent video-assisted thoracic surgeries (VATS), and 30 (26.5%) underwent thoracotomies. Fifty-five patients (48.7%) were preoperatively screened for COVID-19 and received negative results. Fifty-six postoperative complications were reported in 35 patients (30.9%). A prolonged air leak was detected in 11 patients (9.7%), atrial fibrillation in 11 patients (9.7%), empyema in 5 patients (4.4%), pneumonia in 9 patients (7.9%) and lobar atelectasis in 7 patients (6.2%). Three patients (2.7%) with postoperative pulmonary complications required mechanical ventilation, and two of them (1.6%) underwent tracheostomy. Two patients (1.6%) were postoperatively diagnosed as positive for COVID-19. Conclusions: Our data demonstrate the feasibility and efficacy of implementing precautionary strategies to ensure the safety of lung cancer patients undergoing pulmonary resection during the COVID-19 pandemic. The strategy was equally effective in protecting the surgical staff and healthcare providers, and we recommend performing lung cancer surgery during the pandemic era.

## 1. Introduction

The COVID-19 pandemic, caused by the SARS-CoV-2 coronavirus in 2020, is a global crisis that resulted in over millions of deaths worldwide [1]. The SARS-CoV-2 coronavirus was identified in Wuhan, China in December 2019. The first COVID-19 patient in Israel was diagnosed in February 2020. From February to December 2020, in Israel, were two waves of the COVID-19 pandemic, from February to May, and from May to November. From December 2020, vaccination against COVID-19 infection began in Israel. The transmission mechanism of SARS-CoV-2 is primarily by respiratory droplets and direct contact [2]. In general, patients who undergo endotracheal intubation, upper endoscopy, bronchoscopy, surgery on upper airways and bronchopulmonary surgery could potentially transmit COVID-19 infection. We therefore adopted the recommendations of the Centers for Disease Control for the medical staff to use N-95 respirator masks during thoracic surgeries for patients with confirmed COVID-19 [3]. Another challenge we faced was the postponement of elective surgeries due to the overwhelming of healthcare services during the COVID-19 pandemic. However, many hospitals worldwide, despite the shortage of medical and nursing staff during the COVID-19 pandemic, were able to continue to provide care for oncologic patients in order to enable timely treatment [4].

Given that, the surgical treatment of aggressive oncologic diseases, such as non-small cell lung carcinoma (NSCLC), may be time-dependent; each case should be discussed with operating room administrators during the pandemic era as routine practice [5]. The risk–benefit ratio for these patients should be carefully weighed in light of the new reality of risk to contract the COVID-19 virus in the postoperative period, which would lead to higher morbidity and mortality [6].

We hypothesized that operating division activity in the era of the COVID-19 pandemic could be balanced between the decrease in non-oncologic surgeries and giving unique opportunities to provide cancer surgeries. Our study aim and rationale was to present the strategy for, and experience of, providing major lung cancer surgery during the COVID-19 pandemic, while taking special precautionary measures for the safety of the patients and the professional staff. 

## 2. Patients and Methods

We investigated the data of 113 patients with NSCLC who were operated on at Tel Aviv Medical Center, in a large university-affiliated hospital, from February 2020 through December 2020. Data collected from the clinical charts, surgical and pathology reports, and follow-up records were complete for all patients. Inclusion criteria for this observational cross-sectional study: all adult patients with NSCLC that underwent anatomic pulmonary resection, and patients after neoadjuvant chemotherapy or chemoradiation proceeded with major pulmonary resection. All patients that underwent anatomic pulmonary resection for non-NSCLC pathology were excluded from the study.

The selected baseline variables that were assessed included patient demographics, comorbidities, symptoms related to coronavirus infection, preoperative reverse transcription polymerase chain reaction (RT-PCR) testing for the detection of SARS-CoV-2, induction therapy, pathologic (postoperative) stage of the disease, tumor size, location and histology, side and type of surgery, postoperative complications, overall morbidity, mortality and hospital stay. Information on operating room availability, preoperative airway management and postoperative pulmonary complications was also retrieved. 

Approval was obtained from the Institutional Review Board, which waived informed consent for this study (approval number 0927-20-TLV). 

Pulmonary resection was defined as anatomic segmentectomy, lobectomy or pneumonectomy. The standard preoperative evaluation for all NSCLC patients included a complete medical history, a physical examination, a chest X-ray on the day before surgery, as well as electrocardiography, contrast-enhanced computed tomography (CT) of the chest, pulmonary function tests, complete blood counts, chemistry profiles and coagulation tests, positron-emission tomography-computed tomography (PET-CT) and brain imaging for patients with stages II–III NSCLC. According to recommendations of the Israeli Ministry of Health, during the first months of the pandemic in Israel, the preoperative medical history included a questionnaire related to the COVID-19 clinical symptoms. Patients with a positive answer to any of the first six screening questions were referred to SARS-CoV-2 virus detection (RT-PCR) prior to surgery, and a positive answer on question 7 postponed the surgery. From the 5th month of the pandemic, and according to new recommendations, all patients were screened for COVID-19 72 h preoperatively.

The operating rooms were generally available, mostly due to fewer elective non-oncologic surgeries scheduled because of the COVID-19 pandemic and the hospital administration’s decision to continue to operate on all cancer patients, including NSCLC patients.

All patients underwent surgery with the medical staff employing the same, full, personal protective equipment (PPE) used for COVID-19-positive patients: N-95 respirator mask, face shield and full surgical gown. All healthcare providers were screened for COVID-19 every 2 weeks throughout the pandemic period. 

The comorbidities in the study were defined as follows: congestive heart failure defined as a reduced ejection fraction of less than 45%; cardiac comorbidity, defined as the presence of coronary artery disease, any previous cardiac surgery or catheterization, and current cardiac failure or arrhythmia; chronic renal failure (CRF), defined as an elevated creatinine level of >1.5 mg/dL; and chronic obstructive pulmonary disease (COPD), defined as a forced expiratory volume in 1 s/forced vital capacity ratio less than 70%. Tumors were classified and staged according to the eighth edition of the International System for Lung Cancer Staging [7]. The resectability of the lesion and the surgical plan was confirmed at multidisciplinary meetings by the institutional thoracic tumor board team. Cervical mediastinoscopy was performed preoperatively, and bronchoscopy was performed concurrently with resection for patients with enlarged mediastinal lymph nodes (greater than 10 mm) or those with hyper-metabolic fluoro-deoxy-glucose (FDG) uptake in the mediastinal lymph nodes on PET-CT. Hilar and mediastinal lymph nodes were dissected or sampled from at least three mediastinal/hilar nodal sites intraoperatively. Early hospital mortality was defined as death occurring within 30 days of surgery. Standard postoperative care was provided for chest tube management, pain management and anticoagulation parameters [8]. 

## 3. Results

One hundred and thirteen patients underwent major pulmonary anatomic resections due to NSCLC at Tel Aviv Medical Center from February 2020 to December 2020. The answers to the first six items of the preoperative questionnaire on the COVID-19 pandemic were available for all 113 study patients, and the answer to the last question was available for 55 patients (48.7%) (Table 1). The study included 68 males (60.2%) and 45 females (39.8%) with a median age of 68.2 years (range, 41–89). The patients’ demographics, radiologic, histologic and surgical characteristics, comorbidities and postoperative complications are summarized in Table 2.

Before surgery, 14 patients underwent induction therapy due to potentially resectable locally advanced NSCLC (12 underwent chemoradiation and two underwent chemotherapy). Induction therapy protocols did not changed during the COVID-19 pandemic. 

The type of anatomic resections and the pre-induction staging for the locally advanced NSCLC patients is summarized in Table 3. Eighty-three patients (73.5%) underwent video-assisted thoracic surgery procedures. Thirty patients (26.5%) underwent thoracotomy: 5 pneumonectomies, 2 sleeve resections and 23 lobectomies. The reasons for thoracotomy were: nine patients with central tumors, nine after induction therapy, one after previous thoracotomy; one for a superior sulcus tumor (SST); one for a member of the Jehovah Witnesses (JW) congregation; one for a patient with a left upper lobe tumor after coronary artery bypass grafting (CABG); and one converted from VATS due to bleeding.

During the first five months of the pandemic in Israel, fifty-eight patients had not undergone mandatory PR-PCR testing for COVID-19 (51.3%) in accordance with the Ministry of Health’s recommendations, five of whom responded positively to one of the screening questions and were referred for testing. The COVID-19 screening questionnaire is presented in Table 1. Their RT-PCR test results were negative, and they went on to pulmonary resections. Thereafter, all 55 patients were screened 72 h preoperatively for COVID-19 (48.7%) and had negative findings.

Fifty-six postoperative complications were sustained by 35 patients (30.9%). A prolonged air leak (defined as an air leak persisting up to 5 days after the surgery) was detected in 11 patients (9.7%), 11 patients (9.7%) had atrial fibrillation, 5 patients (4.4%) had empyema, 9 patients (7.9%) developed pneumonia and 7 patients (6.2%) developed lobar atelectasis confirmed by chest X-ray, and they underwent urgent bronchoscopy. Three patients (2.7%) with COPD (chronic obstructive pulmonary disease) had respiratory complications and required mechanical ventilation, and two of them (1.6%) underwent tracheostomy. Management of pulmonary complications included tracheobronchial suctioning, nebulizer therapy, high-flow nasal cannula oxygenation, bronchoscopy, tracheostomy and mechanical ventilation, which required the use of full PPE despite the negative RT-PCR tests for COVID-19. These patients, including the three mechanically ventilated patients who underwent a tracheostomy and were managed in the intensive care unit, were screened for the SARS-CoV-2 virus every week, and were found to be negative. Three patients (2.7%) had wound infections; three patients (2.7%) sepsis; and one (0.9%) late post-pneumonectomy broncho-pleural fistula (BPF) that was treated by re-thoracotomy with BPF closure by muscle flap. There was one early (in-hospital) mortality (0.9%) in a patient after pneumonectomy, who developed post-pneumonectomy empyema with septic shock and multi-organ failure. Two patients (1.6%) that were screened 72 h preoperatively for COVID-19 and were found to be negative, were diagnosed postoperatively with positive RT-PCR tests for COVID-19 and treated on an ambulatory basis.

The data on the pathologic staging of the 113 patients are summarized in Table 4. R0 resection (complete resection of a tumor with free margins) was achieved in all 113 patients (100%), including 14 patients after neoadjuvant chemotherapy or chemoradiation. The median number of lymph nodes dissected for all hilar and mediastinal sites was 10.9 (range, 5–23). The median hospital stay for all operated patients was 7.3 days (range, 3–33). One-hundred and twelve patients have completed a follow-up to date, and none of the thoracic team members have tested positive for COVID-19.

## 4. Discussion

In this study, we demonstrate surgical management efficacy and safety in 113 NSCLC patients during the COVID-19 pandemic. The main limitation of this observational cross-sectional study was the fact that we collected data from defined patients with NSCLC and other comorbidities and provided a snapshot of this population at a specific time point. 

Surgical resection with a hilar and mediastinal lymph node dissection is the gold standard for the management of early-stage NSCLC and resectable locally advanced NSCLC down-staged after induction therapy. The American College of Surgeons published recommendations for prioritization of surgical cases during the COVID-19 era [9]. They noted that a delay of over 8 weeks in providing treatment for NSCLC is an independent risk factor for disease progression. They also recommended that one should proceed with elective lung surgery whenever there is hospital capacity and when resources are available [3]. Importantly, Li et al. recommended preoperative screening for SARS-CoV-2 infection, with postponement of lung cancer surgery in positive cases until full recovery is achieved due to high risk for mortality following thoracic surgical procedures [10], and Merritt et al. described increased risks of nosocomial transmission by aero solation of the SARS-CoV-2 virus [3]. N-95 respirator masks are recommended for use by all operating room staff (surgical, anesthesiology, nursing and others) during pulmonary resection in the era of the COVID-19 pandemic, regardless of the patient’s COVID-19 status [11]. 

Because there is no treatment for COVID-19 [1], the only way to reduce perioperative morbidity and mortality related to COVID-19 in patients undergoing surgery for NSCLC is to identify symptomatic COVID-19 patients before surgery and postpone surgery accordingly. In addition, to hasten time to operation, Bilkhu R et al. advised considering performing all elective non-oncologic surgeries in private hospitals, while continuing to operate on patients with cancer in non-private institutions [12]. Kapetanakis El et al. recommended postponing surgery in stage Ia–IIa NSCLC patients for 8 weeks, with new imaging obtained every 6 weeks [13]. Our hospital policy is to provide service for COVID-19 patients while prioritizing medical care for other urgent and cancer patients, including lung cancer patients who require surgical solutions. 

The current protocol in our hospital requires wearing a surgical mask in all clinical areas. Aerosol-generating procedures, such as respiratory suctioning, nebulizer therapy, high-flow nasal cannula oxygenation, bronchoscopy and tracheostomy care, require the use of full PPE. The same standards are followed in the thoracic operating room that, together with the appropriate preoperative assessment PR-PCR testing for COVID-19, enabled us to perform 113 major anatomic pulmonary resections during the COVID-19 pandemic in Israel for NSCLC patients with low mortality (0.9%) and accepted morbidity (30.9%), results that are compatible with the current literature [14].

The Thoracic Surgery Outcomes Research Network recently provided a guide for triaging patients with thoracic malignancies during the COVID-19 pandemic [15]. They defined three phases of hospital status on the prevalence of patients with COVID-19 within the hospital setting: the availability of hospital resources, the prevalence of infections and the consequent resource depletion. Recent publications supported the same considerations on the part of thoracic clinical and surgical teams for the management of lung cancer patients [16,17,18]. The largest reports from Italy and USA evaluated the safety of surgery during the COVID-19 pandemic on 71 and 21 lung resections, respectively, with no reported COVID-19 postoperative infections [19,20]. 

We conclude that our experience demonstrates the feasibility and efficacy of implementing precautionary strategies to ensure the safety of cancer patients undergoing major pulmonary resections during the COVID-19 pandemic in Israel. The strategy was equally effective in protecting the surgical staff and healthcare providers, who continue to undergo periodic PR-PCR testing for COVID-19, and our recommendation is to continue performing lung cancer surgery during the pandemic era.

## Figures and Tables

**Table 1 jcm-11-01102-t001:** COVID-19 screening questionnaire (48–72 h prior to surgery).

1. Are you quarantined according to Ministry of Health instructions?
2. Did you recently arrive from abroad?
3. Did you come from a high-risk (pandemic) zone?
a. rehabilitation center
b. pandemic region
c. densely populated region
4. Have you been in close contact with a person with a positive COVID-19 test?
5. Do you have any of the following symptoms?
a. Fever
b. Cough
c. Throat pain
d. New symptoms of breathing difficulties
e. Other new symptoms
6. Have you tested positive for COVID-19?
7. Availability of screening results during 72 h prior to surgery

**Table 2 jcm-11-01102-t002:** Demographic, comorbidity, radiologic, histologic and surgical characteristics of 113 patients.

Characteristic Underwent PET-CT		No. 113	Note
Comorbidities	Non-insulin-dependent diabetes mellitus	19	
	Hypertension	43	
	Chronic obstructive pulmonary disease	35	
	Chronic renal failure	2	One patient on dialysis
	Peripheral vascular disease	14	
	Chronic atrial fibrillation	8	
	s/p Cerebrovascular accident	11	
	Ischemic heart disease	13	4 s/p CABG, 2 s/p MI, 2 CHF
	Obesity	17	
	Other malignancy	32	6 s/p lung cancer
Symptoms suspicious of COVID-19		5	
Underwent preoperative PCR tests for COVID-19		55	
Histology	Adenocarcinoma	83	
	Squamous cell carcinoma	26	
	Neuroendocrine carcinoma	2	
	Poorly differentiated carcinoma	2	
Mortality		1 (0.9%)	
Complications		35 (30.9%)	
	Atrial fibrillation	11 (9.7%)	
	Pneumonia	9 (7.9%)	
	Air leak	11 (9.7%)	
	Atelectasis	7 (6.2%)	
	Mechanical ventilation	3 (2.7%)	
	Tracheostomy	2 (1.8%)	
	BPF	1 (0.9%)	
	Empyema	5 (3.6%)	
	Wound infection	3 (2.7%)	
	Sepsis	3 (2.7%)	
Smokers		68	
Induction treatment	Neo-adjuvant chemo-radiation	12	
	Neo-adjuvant chemotherapy	2	
Positive margins		0	
Surgery side	Right	65	
	Left	48	

CHF = congestive heart failure; MI = myocardial infarction; CABG = coronary artery bypass grafting.

**Table 3 jcm-11-01102-t003:** The type of anatomic resections for all NSCLC patients.

Note Thoracotomy (9 Central Tumors, 9 s/p Induction Therapy, 1 Completion Lobectomy, 1 SST, 1 JW Patient, 1 Patient after CABG, 1 Converted Due to Bleeding) VATS (2 Extrapleural Lobectomy)	Lobe and Side RUL = 29, RLL = 22, LUL = 25, LLL = 17, RML = 4, RUL/RML = 1	No. Lobectomy 98 (75 VATS, 23 Thoracotomy)	Resection VATS/Thoracotomy Lobectomy
	Lingulectomy = 3, RLL and LLL superior segment = 2, RUL and LUL upper division = 2	Segmentectomy 7	VATS segmentectomy
1 s/p induction therapy	RUL carinal resection = 1, RUL sleeve = −1	Sleeve lobectomy 2	Thoracotomy sleeve lobectomy
Extrapleural-2, intrapericardial-1, 4 s/p induction therapy	Right = 4, left = 1	Pneumonectomy 5	Thoracotomy pneumonectomy
		83	VATS
		30	Thoracotomy
	1	SST-T4N2 bulky disease	Pre-neo-adjuvant staging (14 patients)
	3	T4 diaphragm N2 non-bulky disease	
	2	T4 mediastinal N1	
	1	T4 carinal N1	
	1	T4 mediastinal N0	
	4	T2a/bN2 non-bulky disease	
	2	T3N2 non-bulky disease	

VATS = video-assisted thoracic surgery, CABG = coronary artery bypass grafting, JW = Jehovah Witnesses, SST = superior sulcus tumor, RUL = right upper lobe, RLL = right lower lobe, LUL = left upper lobe, LLL = left lower lobe, RML = right middle lobe.

**Table 4 jcm-11-01102-t004:** Pathologic staging of 113 patients who underwent surgery for NSCLC.

TNM	Surgical Staging (113 Patients)	Staging Post-Neo-Adjuvant (14/113 Patients)
T0N0	9	7 *
T1aN0	10	2 **
T1bN0	34	
T1cN0	19	
T1cN1	1	
T2aN0	10	
T2bN0	8	1 **
T3N0	9	
T4N0	2	1 **
T1aN1	1	
T1bN1	1	1 **
T2aN1	2	
T3N1	1	
T1bN2	1	
T1cN2	1	1 **
T2aN2	3	1 ***
T1bN0M1b	1	

***** patients with complete pathologic response. ** patients down-staged after neo-adjuvant induction therapy. *** patient with skip metastases.

## Data Availability

The data collected and analyses during the current study are available from the corresponding author according to the request.

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
