# Peer review of "Major Pulmonary Resection for Non-Small Cell Lung Carcinoma during the COVID-19 Pandemic—Single Israeli Center Cross-Sectional Study"

_jcm, 2022, doi:10.3390/jcm11041102_

Round 1

Reviewer 1 Report

Dear Editor and Authors,

It was really interesting to read and evaluate this manuscript titled “Major Pulmonary Resection for Non-Small Cell Lung Carcinoma during the COVID-19 Pandemic “ from the Department of Thoracic Surgery at the Tel Aviv Medical Center in Israel.

In this work Dr. Peer and his colleagues present their experience with how they were able to continue performing lung cancer surgery during the pandemic and the methodology and procedures they used to prevent contagion amongst their patients and staff.

As a thoracic surgeon who has struggled to maintain optimal oncological management for his patients during the pandemic and who also has contributed actively in the management and treatment of Covid-19 patients as part of his dual role, this reviewer was very keen to read our fellow surgeons from Israel report and “compare notes” so to speak regarding practices and outcomes.

I refer you to our own approach which was somewhat different to the ones the authors have utilized:

Kapetanakis EI, Tomos IP, Karakatsani A, Koumarianou A, Tomos PI. Management of surgical lung cancer patients during the COVID-19 pandemic in the financially and resource strained Greek health care system. J Surg Oncol. 2020 Aug;122(2):124-127. doi: 10.1002/jso.25988.

Kapetanakis EI, Filippiadis DK, Tomos IP, Karakatsani A, Koumarianou A, Tomos PI. The role of percutaneous interventions in the management of lung cancer patients during the Covid-19 pandemic. J Surg Oncol. 2020 Oct;122(5):989-991. doi: 10.1002/jso.26084.

This is a well written manuscript, quite detailed and with a clear message. The results are as expected and the discussion is well though out. The language is concise and adequate with only minor English language proofreading required.

I would be very keen to seeing this work put out in the literature to demonstrate the feasibility of performing lung cancer surgery during the pandemic with excellent results.

Author Response

Answer:

First of all thank you very much for your comments! I really appreciate it!

We studied and read the following article: Kapetanakis EI, Tomos IP, Karakatsani A, Koumarianou A, Tomos PI. Management of surgical lung cancer patients during the COVID-19 pandemic in the financially and resource strained Greek health care system. J Surg Oncol. 2020 Aug; 122(2):124-127. We addressed the issue in the article in Discussion section and added the Reference 13:

Kapetanakis El et al. recommended to postpone surgery in stage Ia-IIa NSCLC patients for 8 weeks with new imaging obtained every 6 weeks (reference 13 in the manuscript).

We also proofread the article.

Many thanks once more!

Best regards, Michael.

Reviewer 2 Report

In this study, the authors reported they could safely perform major pulmonary resection for 113 non-small cell lung cancer (NSCLC) patients during the COVID-19 pandemic by their implementing precautionary strategy. I strongly agree the necessity of their institutional precautional strategy against the COVID-19 infection, and major pulmonary rejection should be performed under well established screening system such as theirs. I have some comments and concerns.

1. In this study, the authors mainly focused on surgical characteristics, complications and mortality. But when we discussed about the treatment for NSCLC patients during COVID-19 pandemic, I think there are more important things other than surgical outcome. The delay of treatment and others. They should address.
2. In this study, preoperative PCR were tested in just only 55 patients (48.7%). Why just 48.7%? This inclusion criteria seems to be a problem for me. If they think 100% patients underwent major pulmonary resection should receive preoperative PCR test, just 58 patients screened by PCR test should be enrolled in this study. Should address. Because 2 patients postoperatively diagnosed as positive. This is my concern. Didn’t these 2 patients receive PCR test? They did not mention in the main text. Should do.

3. The complications rate was 30.8%. I think this rate is relatively high. Additionally, the rates of critical complications such as mechanical ventilation and tracheostomy are thought to be also high. Why? Should address.

4. From February to December 2020, how was the COVID-19 status in Israel and Tel Aviv? Should mention.

5. Ref 19. was not cited in the main text. Should delete.

Author Response

Reviewer 2

Thank you for your important comments and the opportunity to improve the manuscript!

Comment 1: In this study, the authors mainly focused on surgical characteristics, complications and mortality. But when we discussed about the treatment for NSCLC patients during COVID-19 pandemic, I think there are more important things other than surgical outcome. The delay of treatment and others. They should address.

Answer: I completely agree with the reviewer. The issue about delay of surgical treatment of NSCLC patients in the COVID-19 era is very important. It was the decision of our hospital administration to perform all life-saving surgeries including cancer surgeries during the pandemic in Israel. We added it in the Patients and methods section:

The operating rooms were generally available, mostly due to fewer elective non-oncologic surgeries scheduled because of the COVID-19 pandemic and the hospital administration's decision to continue to operate all cancer patients including NSCLC patients.

Comment 2: In this study, preoperative PCR were tested in just only 55 patients (48.7%). Why just 48.7%?  This inclusion criteria seems to be a problem for me. If they think 100% patients underwent major pulmonary resection should receive preoperative PCR test, just 58 patients screened by PCR test should be enrolled in this study. Should address. Because 2 patients postoperatively diagnosed as positive. This is my concern. Didn’t these 2 patients receive PCR test? They did not mention in the main text. Should do.

Answer: I completely agree with the comment. The policy of screening of surgical patients in our hospital was strongly coordinated with our medical center administration and Israeli Ministry of Health. The recommendations during the first 4-5 months of pandemic in Israel included a questionnaire related to the COVID-19 symptoms. From 5th month of the pandemic in Israel all surgical patients were asked to perform COVID-19 PCR- test 72-hours preoperatively. I explained it in the Patients and methods section:

According to recommendations of the Israeli Ministry of Health, during first months of the pandemic in Israel, the preoperative medical history included a questionnaire related to the COVID-19 clinical symptoms. Patients with a positive answer to any of the first six screening questions were referred to SARS-Cov-2 virus detection (RT-PCR) prior to surgery, and a positive answer on question 7 postponed the surgery.  From the 5th month of the pandemic, and according to new recommendations, all patients were screened for COVID-19 72-hours preoperatively.The 2 patients that were postoperatively diagnosed as positive, were screened 72-hours preoperatively for COVID-19 and found negative. The issue mentioned in Results section:

Two patients (1.6%) that were screened 72-hours preoperatively for COVID-19 and found negative, were diagnosed postoperatively with positive RT-PCR tests for COVID-19 and treated on ambulatory basis.

Comment 3: The complications rate was 30.8%. I think this rate is relatively high. Additionally, the rates of critical complications such as mechanical ventilation and tracheostomy are thought to be also high. Why? Should address.

Answer: The rate of complications is acceptable (14). We added the comment in Discussion section and added the Reference 14:

113 major anatomic pulmonary resections during the COVID-19 pandemic in Israel for NSCLC patients with low mortality (0.9%) and accepted morbidity (30.9%), that are compatible with the current literature [14].

  1. Brunelli A, Rocco G, Szanto Z, Thomas P, Falcoz PE. Morbidity and mortality of lobectomy or pneumonectomy after neoadjuvant treatment: an analysis from the ESTS database. Eur J Cardiothorac Surg. 2020 Apr 1; 57(4):740-746. 35.

Regarding the critical complications (mechanical ventilation, tracheostomy) and others, they generally occurred in COPD patients and I added the comment about it in the Results section:
Three patients (2.7%) with severe COPD (chronic obstructive pulmonary disease) had respiratory complications and required mechanical ventilation, and two of them (1.6%) underwent tracheostomy.

Comment 4: From February to December 2020, how was the COVID-19 status in Israel and Tel Aviv? Should mention.

Answer: From February to December 2020, in Israel and Tel-Aviv, were 2 waves of COVID-19 pandemic. The first COVID-19 patient in Israel was diagnosed in February 2020. The first wave: February to May 2020: on 25 March, the government imposed stricter restrictions on citizens' movements ("The Government Approved Emergency Regulations to Restrict Activities in Order to Curb the Spread of Coronavirus in Israel". Ministry of Health. 25 March 2020. https://www.gov.il/en/departments/news/25032020_01). Second wave: May to November 2020. The government announced new social distancing guidelines ("Government orders closure of event halls, culture venues, gyms and nightclubs". The Times of Israel. 5 July 2020). From December 2020, vaccination against COVID-19 infection began in Israel.

We mentioned the issue in the Introduction section.

Comment 5: Ref 19. was not cited in the main text. Should delete.

Answer: Thank you. Corrections have been made

Many thanks once more!

Best regards, Michael.

Round 2

Reviewer 2 Report

The authors have revised the manuscript and addressed the Reviewer's comments.